# Methemoglobinemia, Increased Deformability and Reduced Membrane Stability of Red Blood Cells in a Cat with a *CYB5R3* Splice Defect

**DOI:** 10.3390/cells12070991

**Published:** 2023-03-24

**Authors:** Sophia Jenni, Odette Ludwig-Peisker, Vidhya Jagannathan, Sandra Lapsina, Martina Stirn, Regina Hofmann-Lehmann, Nikolay Bogdanov, Nelli Schetle, Urs Giger, Tosso Leeb, Anna Bogdanova

**Affiliations:** 1Clinic for Small Animal Internal Medicine, Vetsuisse Faculty, University of Zurich, 8057 Zurich, Switzerland; 2Institute of Genetics, Vetsuisse Faculty, University of Bern, 3001 Bern, Switzerland; 3Clinical Laboratory, Department of Clinical Diagnostics and Services, and Center for Clinical Studies, Vetsuisse Faculty, University of Zurich, 8057 Zurich, Switzerland; 4Red Blood Cell Group, Institute of Veterinary Physiology and the Center for Clinical Studies, Vetsuisse Faculty Zurich, University of Zurich, 8057 Zurich, Switzerland

**Keywords:** feline, hematology, red blood cells, cyanosis, erythrocytosis, microcytosis, splicing, precision medicine

## Abstract

Methemoglobinemia is an acquired or inherited condition resulting from oxidative stress or dysfunction of the NADH-cytochrome b5 reductase or associated pathways. This study describes the clinical, pathophysiological, and molecular genetic features of a cat with hereditary methemoglobinemia. Whole genome sequencing and mRNA transcript analyses were performed in affected and control cats. Co-oximetry, ektacytometry, Ellman’s assay for reduced glutathione concentrations, and CYB5R activity were assessed. A young adult European domestic shorthair cat decompensated at induction of anesthesia and was found to have persistent methemoglobinemia of 39 ± 8% (reference range < 3%) of total hemoglobin which could be reversed upon intravenous methylene blue injection. The erythrocytic CYB5R activity was 20 ± 6% of normal. Genetic analyses revealed a single homozygous base exchange at the beginning of intron 3 of the *CYB5R3* gene, c.226+5G>A. Subsequent mRNA studies confirmed a splice defect and demonstrated expression of two mutant *CYB5R3* transcripts. Erythrocytic glutathione levels were twice that of controls. Mild microcytosis, echinocytes, and multiple Ca^2+^-filled vesicles were found in the affected cat. Erythrocytes were unstable at high osmolarities although highly deformable as follows from the changes in elongation index and maximal-tolerated osmolarity. Clinicopathological presentation of this cat was similar to other cats with CYB5R3 deficiency. We found that methemoglobinemia is associated with an increase in red blood cell fragility and deformability, glutathione overload, and morphological alterations typical for stress erythropoiesis.

## 1. Introduction

Heme groups in hemoglobin can only transport oxygen when containing iron in the ferrous state (Fe^2+^). Oxidation of Fe^2+^ to Fe^3+^ converts hemoglobin to methemoglobin (metHb), which is unable to bind and transport oxygen [1]. Reduction of Fe^3+^ to Fe^2+^ is mainly catalyzed by the erythrocytic enzyme cytochrome b5 reductase 3 (CYB5R3) a. Impaired hemoglobin reduction results in increased blood methemoglobin levels (>3% metHb) [2]. Methemoglobinemia can be acquired due to exposure to oxidative agents [2] or hereditary [3]. Hereditary methemoglobinemia is most commonly due to diminished erythrocytic CYB5R activity, while low cytochrome b5 levels, abnormal hemoglobin M, or defects compromising the production of antioxidants are less common causes [1,3,4,5]. Persistent methemoglobinemia may also be caused by flavin adenine dinucleotide (FAD) deficiency [6]. There are soluble (mostly erythrocytic) and membrane-bound ubiquitous CYB5R3 isoforms [1,5,7]. In humans, erythrocytic CYB5R3 deficiency is characterized by cyanosis and mild clinical signs and referred to as type I recessive congenital methemoglobinemia (type I RCM; Online Mendelian Inheritance in Man [OMIM] #250800) [8,9]. In contrast, human patients with membrane-bound CYB5R3 deficiency, referred to as type II RCM, exhibit developmental abnormalities and neurologic impairments (Type II RCM: OMIM #250800) [10,11]. More than 80 different pathogenic variants in the *CYB5R3* gene and their correlation with either type I or type II RCM have been reported in human patients [5,7,12,13].

Naturally occurring CYB5R3 deficiency causing methemoglobinemia (equivalent to type I RCM in humans) has also been reported in dogs (Online Mendelian Inheritance in Animals [OMIA] #002131-9615, [14,15,16,17,18,19,20,21]) and cats (OMIA #002131-9685; #01171-9685, [14,15,22,23,24,25]). While clinicopathological manifestations of hereditary methemoglobinemia, such as cyanosis, are well-described in humans [3,5] and in animals [6,14,18,19,22,26], the genotype-phenotype correlation has not been well documented. The impact of high metHb levels on red blood cell (RBC) properties were not studied in humans or in animals, albeit Heinz bodies have been described in a single affected cat [6,14,18,19,22,26]. Pyknocytosis and eccentrocytosis observed in horses with methemoglobinemia caused by FAD deficiency [6] was never reported in cats or dogs with reduced erythrocytic CYB5R3 activity. While the diminished oxygen transport capacity of blood in patients and the affected animals with CYB5R3 deficiency is recognized, and modest erythrocytosis is described for some of them [9,14], RBC hydration, deformability, and stability have not been characterized.

In this report, we describe the clinicopathological manifestations and RBC abnormalities in a domestic shorthair cat with methemoglobinemia likely caused by a pathogenic *CYB5R3* splice variant.

## 2. Materials and Methods

### 2.1. Animals, Samples, and Clinical Examination

A European domestic shorthair cat, presented to the Small Animal Clinic of the University of Zurich, Switzerland, was found to have methemoglobinemia. Routine physical and cardiological examinations, as well as hematological and serum chemistry analyses, were applied on several occasions to diagnose and manage this cat’s illness. Leftover K_2_-EDTA anticoagulated blood samples were further examined to determine the degree and cause of the methemoglobinemia and the underlying molecular genetic defect. The erythrocytic activity of CYB5R activity, as well as RBC morphology, rheology, ion/water balance, and reduced thiol content, were also assessed.

Leftover K_2_-EDTA anticoagulated blood samples from five domestic cats, seen at the same clinic for wellness and other reasons, served as controls. The use of leftover samples was approved, and owner consent was received (the Cantonal Committee for Animal Experiments Bern; permit BE 71/19). For the whole genome sequencing analysis, data from 74 additional genetically diverse control cats from the Vetsuisse Biobank were available.

### 2.2. Blood Test Analyses 

Clinical hematology, including complete blood count, were performed using a Cobas 6000 (Roche Diagnostics, Rotkreuz, Switzerland) and a Sysmex XN-1000V (Sysmex Suisse AG, Horgen, Switzerland, respectively. In addition, total hemoglobin (Hb), oxyhemoglobin (O_2_-Hb), carboxyhemoglobin (CO-Hb), and metHb were measured in K_2_-EDTA anticoagulated blood samples using a point-of-care instrument (RAPIDPoint, Siemens, Zurich, Switzerland) and a portable CO-oximeter (Oximeter Avoximeter 4000, Werfen Life Group, Barcelona, Spain). Erythrocytic CYB5R activities were measured at 575 nm using Lambda 25 spectrophotometer (Perkin Elmer, Waltham, MA, USA) with and without NADH according to Hegesh et al. [27] and Beutler [28]. Erythrocytic reduced glutathione (GSH) concentrations were measured as the amount of thionitrobenzoic acid (TNB) liberated from dithionitrobenzoate (DTNB) at 412 nm (Perkin Elmer, Rodgau Germany) according to Ellman’s assay [29,30]. 

For advanced morphological examination of RBCs and intracellular free Ca^2+^ monitoring, an Axiovert 200 M fluorescence microscope (λex = 488 nm; equipped with Zeiss Zen Blue v. 3.1 software, Carl Zeiss, Jena, Germany) and a digital camera (Orca-flash 4.0, Hamamatsu, Solothurn, Switzerland) were used. The RBCs were loaded with 10 µM fluo-4 AM (Thermo Fisher Scientific, Waltham, MA, USA) for 1 h at room temperature in darkness and used for microscopic examination of Ca^2+^ distribution in RBCs. 

Measurements of erythrocyte hydration, osmotic resistance and deformability, and intracellular cation content were performed using the osmoscan mode of Lorrca Maxsis ektacytometry (RR Mechatronics, Zwaag, The Netherlands) [31]. The measured parameters included osmotic tolerance (area), deformability as maximal elongation index (EI_max_) at the osmolarity at which it is reached (O_EImax_), erythrocytic hydration state as the maximally tolerated hyperosmotic osmolarity (O_hyper_), and elongation index at this osmolarity (EI_hyper_), as well as minimal tolerated osmolarity representing osmotic stability of the membrane (O_min_) and the elongation index at this minimal osmolarity (EI_min_). Erythrocytic Na^+^ and K^+^ contents were detected in RBC pellets using flame photometry (Sherwood, Cambridge, UK).

### 2.3. Molecular Genetic Studies and Protein Structure Modeling

Genomic DNA was extracted from K_2_-EDTA anticoagulated blood samples using the Maxwell Instrument and DNA purification kit (Promega, Dübendorf, Switzerland). The isolated genomic DNA was stored at −20 °C until further analyses. In addition, mRNA from freshly collected blood of the affected and one healthy control cat was extracted using the PAXgene RNA kit (PreAnalytiX GmbH, Hombrechtikon, Switzerland) and stored at −80 °C for further analyses. 

An Illumina TruSeq PCR-free DNA library with ~400 bp insert size was prepared from the affected cat, and a total of 197 million 2 × 150 bp paired-end reads were obtained on a NovaSeq 6000 instrument (Illumina, Zurich, Switzerland) representing a 23× coverage. Mapping and alignment to the F.catus_Fca126_mat1.0 reference genome assembly, as well as variant calling, were performed as previously described [32]. The sequenced data were deposited under the study accession PRJEB7401 and the sample accession SAMEA14502950 at the European Nucleotide Archive. Numbering within the feline *CYB5R3* gene corresponds to the NCBI RefSeq accession numbers XM_045062469.1 (mRNA) and XP_044918404.1 (protein).

The SnpEff v. 5.0 software [33], together with NCBI annotation release 105 for the F.catus_Fca126_mat1.0 reference genome assembly, was used to predict the impact of the previously called variants (high, moderate, low, and modifier). For variant filtering, 74 control genomes derived from 44 purebred cats of 13 different breeds and 30 random-bred domestic cats (Appendix A) were available. While not specifically examined, there were no reports of methemoglobinemia in these control cats.

The candidate variant discovered by genome analysis was Sanger sequenced and genotyped in affected and control cats. To amplify the extracted genomic DNA, AmpliTaq Gold 360 Mastermix (Thermo Fisher Scientific, Waltham, MA, USA) together with the primers 5′-TGT GTG CAG CTT GTG AGT CC-3′ (Primer F) and 5′-CAG CCA CAC CTT TGC TCA C-3′ (Primer R) were used. After an initial denaturation step of 10 min at 95 °C, 30 cycles of 30 s denaturation at 95 °C, 30 s annealing at 60 °C, and 1 min of polymerization at 72 °C, followed by an extension step of 7 min at 72 °C and cooling the samples down to 4 °C for storage. The isolated RNA was reverse transcribed using SuperScript^®^ IV Reverse Transcriptase Kit (Thermo Fisher Scientific). The generated cDNA was then amplified with AmpliTaq Gold 360 Mastermix (Thermo Fisher Scientific) using the forward primer 5′-ACA TCA AGT ACC CGC TGA GG-3′ located in exon 2 and a reverse primer 5′-CAC CGT GTA CCA GAG CTT GA-3′ located in exon 8 with the PCR conditions as described above.

Quality control and fragment length analyses of both gDNA and cDNA PCR products were performed using a 5200 Fragment Analyzer (Agilent, Santa Clara, CA, USA), treating the samples with exonuclease I and alkaline phosphatase. PCR amplicons were sequenced on an automated sequencer (ABI 3730 DNA Analyzer, Thermo Fisher Scientific) and analyzed with the Sequencher 5.1 software (Gene-Codes, Ann Arbor, MI, USA). 

The ColabFold protein modeling tool [34] was used to generate 3D models of the wild-type and mutant CYB5R3 proteins and predict translation products of the aberrant transcripts (mut #1 and mut #2). The FASTA sequence of accession XP_044918404.1 was used for the wild-type CYB5R3 protein.

## 3. Results

### 3.1. Clinical Examination and Blood Analyses

A two-year-old female spayed European domestic shorthair cat was presented to the Small Animal Clinic of the University of Zurich for surgical fracture repair. During induction of anesthesia for surgical repair, the cat decompensated and was found to be cyanotic. Venous blood gas analyses with a point-of-care instrument showed reduced oxygen saturation and were unable to detect metHb. In a clinical “metHb spot-test”, the cat’s blood remained persistently brown compared to normal venous blood, which turned bright red after exposure to air (Appendix A). Severe methemoglobinemia was detected using a portable laboratory CO-oximeter (Table 1). Routine blood tests revealed mild microcytosis (Table 1), and chest radiographs and echocardiography readouts were within the reference range limits.

Due to the severe methemoglobinemia and lack of response to 100% oxygen, therapy with reducing agents was started, including N-acetylcysteine (initially 140 mg/kg then continued with 70 mg/kg for further seven administrations every six hours [Fluimucil 20%, Zambon, Switzerland], ascorbic acid (30 mg/kg, intravenously daily [Vitamin C 10%, Streuli, Switzerland] and orally S-adenosyl methionine (75 mg, Samylin Small Breed SAMe [VetPlus Ltd., Lytham, UK]). Since these interventions seemed hardly effective (Table 1), 1 mg/kg methylene blue (methylthioninum chloride, Proveblue 5 mg/mL, Provepharm, SAS, Marseilles, France) was administered intravenously which immediately and completely reversed the methemoglobinemia (Table 1). The surgery could be successfully performed, and the cat had an uneventful recovery and has been doing well over the past year without any further treatment. However, routine blood tests continued to show marked methemoglobinemia and mild microcytosis as well as developing erythrocytosis (Table 1).

### 3.2. Erythrocytic Reduced Glutathione Concentration and CYB5R Activity

At each time point, erythrocytic GSH levels of the affected cat were almost double that compared to control cats (Table 1). As no triggers for methemoglobinemia were identified and the therapy with reducing agents was unsuccessful, a hereditary MetHb was suspected. The erythrocytic CYB5R activity measured on two occasions was low compared to control cats (Table 1).

### 3.3. Erythrocyte Morphology, Deformability, and Ion Content

At the time of anesthesia induction, slightly low MCV, high MCHC values, and several echinocytes were noted (Table 1, Figure 1A–D). Microcytosis and echinocytosis persisted during the follow-up visits (Table 1). During steady-state conditions of the affected cat, osmoscan revealed persistently decreased tolerance of RBCs to hyperosmotic stress (decreased hyperosmotic threshold, O_hyper_) compared to the RBCs of controls, but increased deformability (higher elongation index, EI) at both physiological and high osmolarity conditions. The lower O_hyper_ for the affected cat’s RBCs suggested dehydration which was also reflected by mild microcytosis (Figure 1D). Intracellular cation content of RBC did not differ between the affected and control cats (Table 2).

Staining of RBCs with a fluorescent dye fluo-4 AM was used to detect Ca^2+^ distribution within RBCs. In control cats, multiple RBCs contained single vesicles filled with Ca^2+^, while bulk cytosolic Ca^2+^ levels were low (Figure 1E). Some of the RBCs of the affected cat (both discocytes and echinocytes) had high cytosolic Ca^2+^ levels (Figure 1F–H). In some of these Ca^2+^-overloaded cells, no Ca^2+^-filled vesicles were visible (Figure 1H), while the others contained one or multiple Ca^2+^-filled vesicles (Figure 1F–H). The latter was present only in the sample drawn from the affected cat at the first visit to the hospital when anemia associated with fracture was diagnosed. As blood hemoglobin levels normalized in cats with methemoglobinemia, this type of cells was no longer observed (Figure 1F,G).

### 3.4. Genetic Analysis

To identify the causative genetic variant, the genome of the affected cat was sequenced. Since we did not have access to any clinical history of the cat’s parents, we searched for both homozygous and heterozygous private variants in the affected cat compared to the genomes of 74 control cats (Table 3 and Appendix A). Based on the clinical phenotype, we focused our search on private variants in the main candidate gene *CYB5R3* [7]. 

We identified one homozygous private variant located in a splice region of the *CYB5R3* gene. The variant, XM_045062469.1:c.226+5G>A, is a single nucleotide exchange at the beginning of intron 3. On the genomic level, this variant can be designated as ChrB4:135,605,715C>T (F.catus_Fca126_mat1.0 assembly).

The presence of the *CYB5R3* splice region variant was confirmed by Sanger sequencing on genomic PCR products (Figure 2). While a healthy control cat was homozygous for the wildtype genotype (c.226+5G), the affected cat was homozygous for the mutant allele (c.226+5A).

As the genomic variant did not directly affect the canonical GT-dinucleotide at the 5′-splice site of intron 3, we experimentally assessed the consequences of the deletion on the transcript level. RNA derived from the blood of the affected cat and a control cat was reverse transcribed, and the cDNAs were amplified using primers located in exon 2 and exon 8 of the *CYB5R3* gene. The affected cat expressed at least two different transcripts (Figure 3A). One of them, designated mut #1, gave rise to a longer RT-PCR product than the control cat, while the other one, mut #2, was shorter than the control.

Sequence analysis revealed that mut #1 contained an insertion of 36 bp derived from the beginning of intron 3, XM_045062469.1:r.226_227insGTGAGCGCAGCCCTGACCCA-GCCCGAGTGGAACCGG (Figure 3B). Mut #2 lacked 73 nucleotides comprising the entire exon 3, XM_045062469.1:r.154_226del73 (Figure 3B).

### 3.5. Protein Modelling 

In mut #1, the 36 bp insertion maintained the reading frame but is predicted to insert 12 amino acids into the protein XP_044918404.1:p.(G76_Q77insERSPDPARVEPG). The inserted amino acids are predicted to form an additional loop between the FAD and NADH binding domains (Figure 4). In mut #2, the exon skipping results in an early premature stop codon, XP_044918404.1:p.(V52Afs*58), truncating 64% of the wildtype open reading frame.

## 4. Discussion

The *CYB5R3* gene gives rise to a cytosolic/soluble erythrocytic and a membrane-bound ubiquitous CYB5R3 isoform by alternative splicing of exon 1. In humans, pathogenic *CYB5R3* variants may result in two different types of autosomal recessive CYB5R3 deficiency, also referred to as type I and II RCM [5,8]. Type I RCM involves methemoglobinemia and mild clinical signs due to a partial loss of function of the erythrocytic enzyme [3]. In contrast, type II RCM, characterized by the complete or almost complete loss of activity of the CYB5R3 enzymes leads to devastating developmental and neurological disease in humans, likely due to fatty acid disturbances besides methemoglobinemia [10,11].

In animals, only cases of type I and not type II RCM have been described, including methemoglobinemic dogs and cats with *CYB5R3* variants [14]. We report here on another mostly asymptomatic methemoglobinemic cat with severely reduced erythrocytic CYB5R activity and a likely pathogenic *CYB5R3* splice defect causing type I RCM. The CYB5R3-deficient erythrocytes of the affected cat appeared microcytic and presented with less stable but more deformable erythrocytic membranes, while blood GSH levels were markedly elevated. Further studies will be needed to directly associate these features associated with methemoglobin accumulation in the cells due to the CYB5R3 deficiency. 

The affected cat’s *CYB5R3*:c.226+5G>A splice variant results in a larger (mut #1) and a smaller (mut #2) aberrant mRNA transcript. The translation product of mut #1 is predicted to add 12 amino acids to the flavin adenine diphosphate (FAD)-binding domain and may exhibit enzymatic activity. The mut #2 transcript, harboring a frameshift and premature stop codon, was only detectable in minor amounts, possibly due to nonsense-mediated mRNA decay [37]. Another *CYB5R3* splice defect affecting the last base of intron 3 was described in a prior feline case report [23], causing clinical features of the type I RCM rather than the predicted type II RCM seen with truncating variants in humans [7]. 

More than 80 *CYB5R3* variants have been reported in human patients with RCM [5,7,12,13], but a comparable splice defect, as described herein, has not yet been reported in human patients. Humans with type I RCM commonly have missense *CYB5R3* variants in exon 2–9, while those with type II frequently have *CYB5R3* variants leading to protein truncations [7]. While a few *CYB5R3* variants have been reported in families, none of them seem to be prevalent among any human populations. 

Four different missense variants in the *CYB5R3* gene have been reported to be causative for type I RCM in dogs [18,20,21,38]. The *CYB5R3* missense variant p.I194L was found in several Pomeranians worldwide, while the p.R219P missense variant was seen in several Pit Bull terriers in North America; however, their actual prevalence remains to be determined [18]. The other two *CYB5R3* variants, p.G76S and p.T202A, were each only identified in one methemoglobinemic dog. 

Interestingly, in the three species in which type I RCM has been reported as well as in human patients with type I and II RCM, considerable residual erythrocytic CRB5R activities were found, e.g., 10% for dogs (median, IQR 5.0–24.1%, [18]) and <15% for affected cats compared to species–specific controls [23]. It is possible that proteins encoded by the aberrantly spliced transcripts and proteins with specific amino acid exchanges support residual enzyme activity. In the cat described here as well as all other studies in methemoglobinemic humans and animals, erythrocytic activities were measured in vitro by standard methods after cellulose-cellulose column separation to remove leukocytes and platelets, but contamination by the erythrocytic hemolysates with CYB5R isoforms present in other blood cells (white blood cells, platelets) cannot be excluded and may have contributed to the measured enzyme activities in this and other cases. Furthermore, there are other pathways which may contribute to the reduction of methemoglobin to oxyhemoglobin. Thus, it remains unclear if the measured values were truly related to specific erythrocytic CYB5R or non-enzymatic activities. Specific investigations of cytosolic versus membrane-bound CYB5R isozyme protein expression and activities in healthy and methemoglobinemic companions and other animals are needed.

Hereditary methemoglobinemia due to CYB5R3 deficiency is often incidentally diagnosed after recognizing cyanotic mucous membranes. The cyanosis can be marked or only mild, despite erythrocytic methemoglobin concentrations being persistently ~40%. It was a bone fracture that brought the affected cat of this report to the clinic, where the cyanosis was recognized during induction of anesthesia. Likewise, severe methemoglobinemia has been reported in CYB5R3-deficient dogs [14,15,16,17,18,20,21,38] and humans [5,9]. However, while the increase in methemoglobin levels above 50% may lead to seizures and even death in human patients [9], dogs, and cats with CYB5R3 deficiency appear more tolerant to such a massive loss of O_2_ carrying capacity [14,18,22].

Methemoglobinemia typically causes the pulse oximeter to report a saturation of ~80–90% even with oxygen supplementation and high PaO_2_ (e.g., [9,20]). Moreover, diffusion of O_2_ from the alveoli of the lungs to blood plasma is the main determinant of pO_2_ which is not impaired by methemoglobinemia [9]. Thus, despite severe cyanosis, pO_2_ remains normal in individuals with methemoglobinemia. Point-of-care pulse oximetry in patients with methemoglobinemia is therefore not accurate because methemoglobin absorbs both infrared and red light equally, which interferes with the measured percentage of oxyHb and deoxyHb (Hb oxygen saturation SO_2_ of ~86% in our affected cat measured at the time of anesthesia induction). This dissociation of SO_2_ and cyanosis in patients with methemoglobinemia is interpreted as “refractory hypoxemia” and is diagnostically useful [9]. Administrating oxygen raises the pO_2_ but fails to correct cyanosis in patients with methemoglobinemia. In contrast to standard pulse oximetry, a co-oximeter measures light absorbance at four different wavelengths of which 630 nm wavelength is used for accurate determination of metHb. Reliable detection on metHb can also be achieved with laboratory methods recording a full hemoglobin spectrum [39]. In the cat with methemoglobinemia presented herein, the co-oximeter clearly detected severe methemoglobinemia (Table 1).

While methemoglobin concentrations are regularly reported in humans and animals with RCM, little is known about GSH levels. Blood GSH levels in the affected cat were double that of the control cats without hematological disorders. The molecular mechanism behind this finding as well as its relation to metHb accumulation, is currently unknown. Reduced activity of CYB5R protein results in lower NADH consumption, which in turn may influence the ATP-dependent production and maintenance of the GSH pool in the cells. Alternatively, metHb may stimulate glycolysis in RBCs by replacing the glycolytic enzymes at the cytosolic domain of the band 3 protein [40]. If the latter is true, the increase in GSH should only be observed in severe cases of methemoglobinemia when metHb content is comparable to that of deoxyHb. Interestingly, a similar high GSH concentration was reported in one of three CYB5R3-deficient Pomeranian dogs with severe disease manifestation [20]. Whether this GSH overload is indeed compensatory by facilitating antioxidative defense or inducing so-called “reductive stress” due to progressive auto-oxidation, generation of glutathionyl radicals [41], and extended S-glutathionylation of numerous proteins [42,43,44,45] remains to be explored.

Erythrocytic morphological features have not received much attention in humans presented with hereditary methemoglobinemia and animals with CYB5R3 deficiency. The presence of single enlarged cells with irregular immature morphology and a high number of Ca^2+^-filled vesicles (Figure 1C,F,G), as well as the presence of cells with high levels of Ca^2+^ (Figure 1H), is typical for human patients with stress erythropoiesis [46,47,48]. A similar increase in Ca^2+^-filled vesicles was earlier suggested as a sign of young RBCs in humans and horses [49,50].

The investigation of RBCs of the methemoglobinemic cat in this report provides a first insight into their morphology and deformability (Table 3). The bulk population of RBCs was microcytic and more fragile in a hypoosmotic environment. These abnormalities could not be explained by the changes in ion and water content, as erythrocytic Na^+^ and K^+^ concentrations in the RBC of the affected cat were within the reference range obtained for the control animals (Table 2) that are known to have high Na^+^ and low K^+^ erythrocytes [51]. Thus, reduced tolerance to hyperosomotic conditions could not be explained by the dehydration of RBCs of methemoglobinemic cat. In humans and mice, deoxyHb interacts with higher affinity than oxyHb with its docking site on the cytosolic band 3 protein domain [52,53,54]. Similar to deoxyHb, metHb [40] and hemichrome complexes [55] were also shown to interact with high affinity with the same docking site at the band 3 protein. The band 3 protein is one of the most ubiquitous transmembrane proteins in the RBC membrane. The band 3 protein joins the other transmembrane proteins, forming the Ankyrin and Junctional complexes to which the horizontal spectrin cytoskeletal network is attached [56]. The binding of deoxyHb or metHb to the cytosolic domain of the band 3 protein causes the detachment of ankyrin and the spectrin mesh from band 3 [57]. As a result, the membranes of deoxygenated RBCs become less rigid and more deformable but also more fragile [58]. Similar but more severe changes in the cytoskeletal structure are observed in patients with hereditary spherocytosis caused by mutations in the band 3 protein, ankyrin, or spectrins [56]. The same mechanism may also be implicated in the cat with methemoglobinemia presented in this study. If supported by experimental findings, this hypothesis could also explain microcytosis by the membrane loss due to the destabilization of the RBC membranes caused by metHb interaction with the band 3 protein [40]. Excess Ca^2+^ in the cytosol of some cells of the affected cat would further support membrane loss [59]. It remains unknown how much the microcytosis and increased RBC deformability of CYB5R3-deficient cats affects the O_2_ transport function of these cells in vivo.

Similar to the previously reported human patients as well as animals with CYB5R3 deficiency [5,14], this affected cat remained asymptomatic but was developing mild erythrocytosis due to the persistently severe methemoglobinemia and associated hypoxia (Table 2). However, trauma, the associated inflammation, blood loss, as well as a further decrease in Hb oxygen saturation during anesthesia made the affected cat unstable and required correction of metHb levels. Administration of methylene blue, as the antidote of choice, reduced metHb abundance back to normal in the affected cat (Table 1), as it was reported earlier for humans and animals with CYB5R3 deficiency [17,24,60]. However, because CYB5R3-deficient humans and companion animals are mostly asymptomatic, there is no need for chronic treatment with oral methylene blue or other agents.

In conclusion, the CYB5R3-deficient cat of this report shows features of type I rather than type II RCM, despite having a *CYB5R3* splice defect. Furthermore, this feline CYB5R3-deficient case illustrates the associated changes in RBC morphology and rheology, which have not been investigated in humans or animals with type I RCM.

## Figures and Tables

**Figure 1 cells-12-00991-f001:**
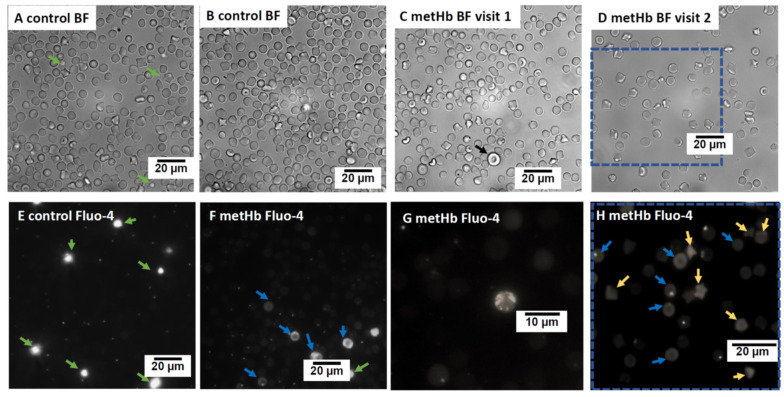
Microfluorescent live cell imaging of RBCs of the affected (metHb) and control cats. Bright field (BF) images of RBCs and Ca^2+^ distribution (using fluorescent signal of Fluo-4 dye) from control and the affected cat. BF images of control (**A**,**B**) and the affected cat at first visit when the affected cat was anemic (**C**) and at second visit when in steady state (**D**). Panels E-H represent the Ca^2+^-dependent fluo-4 fluorescent signal for the RBCs of the control (**E**) and the affected (**F**–**H**) cats. Blue dotted frame indicates the BF and fluorescent images of the same area. Black arrow in panel C highlights an immature large RBC with distorted discoid shape. Platelets (**E**,**F**) are filled with Ca^2+^ and intensely stained (green arrows). Some of the discocytes (blue arrows, (**F**,**H**)) and echinocytes (yellow arrows, (**H**)) of the affected cat were overloaded with Ca^2+^. A few Ca^2+^-overloaded cells of the affected cat contained multiple Ca^2+^-filled vesicles (**F**,**G**), which are characteristic for immature reticulocytes.

**Figure 2 cells-12-00991-f002:**
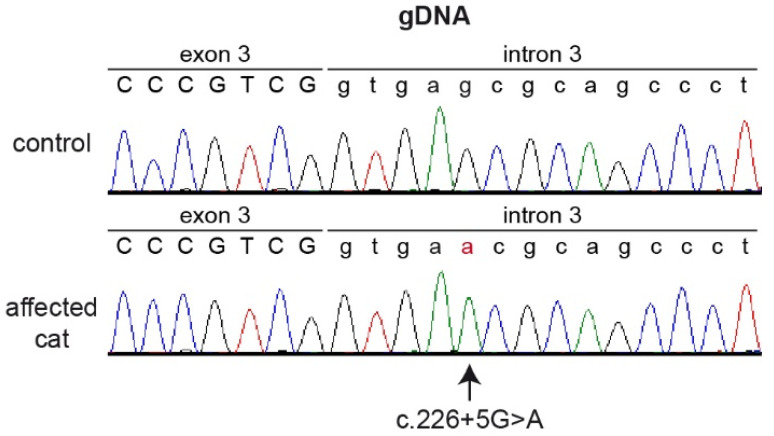
Sanger sequencing electropherograms of the affected and a control cat illustrating the single nucleotide exchange at the beginning of intron 3. The variant position is indicated by an arrow. The altered base is indicated in red.

**Figure 3 cells-12-00991-f003:**
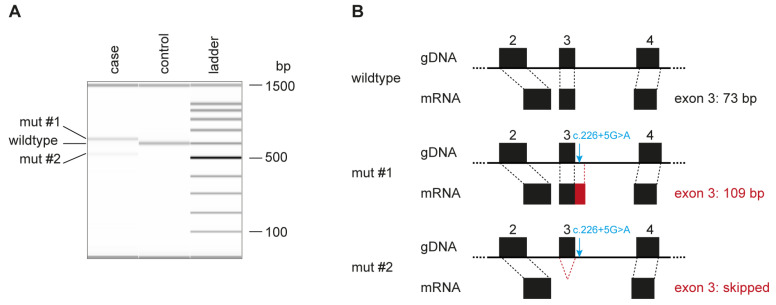
Aberrant *CYB5R3* transcripts in the affected cat. (**A**) Fragment Analyzer bands of the RT–PCR products in the control animal show the expected 599 bp band, while in the affected animal, two different bands are visible (mut #1 and mut #2). Sanger sequencing yielded a length of 635 bp for mut #1 and 526 bp for mut #2. (**B**) Schematic representation of exons 2 to 4 of the *CYB5R3* gene. The c.226+5G>A variant is indicated on the genomic DNA level with a blue arrow. The variant leads to aberrant splicing, adding 36 bp of intron 3 to the mut #1 mRNA and skipping the entire exon 3 in the mut #2 transcript.

**Figure 4 cells-12-00991-f004:**
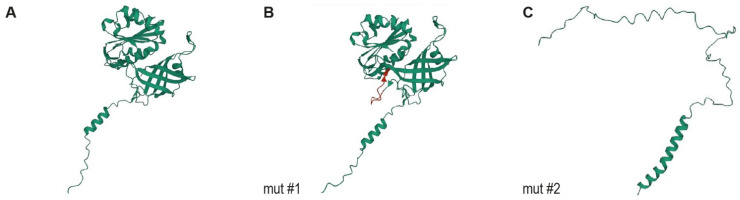
Protein modeling of CYB5R3. (**A**) Protein model of the wildtype feline CYB5R3 protein. (**B**) Protein model of the predicted translation product of transcript mut #1 (XP_044918404.1:p.G76_Q77insERSPDPARVEPG). The 12 inserted amino acids are marked in red. (**C**) Protein model of the predicted translation product of transcript mut #2 XP_044918404.1:p.(V52Afs*58). We did not experimentally verify whether the predicted proteins were expressed. However, the RT–PCR experiments confirmed that the mut #2 mRNA transcript was present in the affected cat, suggesting that at least a fraction of this aberrant transcript escaped nonsense-mediated mRNA decay.

**Table 1 cells-12-00991-t001:** Venous blood gases, erythrocytic metHb, reduced glutathione concentrations, and erythrocytic CYB5R activity in a methemoglobinemic cat.

Parameters	At Time of Anesthesia Induction	After Supplementation *	After Methylene Blue **	After 2 Weeks ***	After ~4 Months ***	After ~6 Months ***	Reference Interval ****
Hematocrit, %	29	33			46	45	33–45
Hemoglobin, g/L	116	121			160		113–155
MCV, fL	36	40			39		40–48
MCHC, g/dL	40	37			35		33–36
Reticulocytes, 10^6^/L					14,076		<45,000
O_2_-Hb,%	23.7	64.5	91.1	47.1	63.4	60.4	>90% #
CO-Hb, %	4.4	2.7	3.6	1.7	2.8	2.2	<1% #
metHb, %	49.6	39.9	3.1	32.4	33.8	35.2	0.3–4.0
GSH, µmol/g Hb	11.9			10.1		9.4	4.8 ± 0.9
CYB5R activity,					0.27	0.46 ± 0.10	2.07 ± 0.81
IU/g Hb					13.0	22.2	100
%							

* Supplement oxygen therapy to reduce methemoglobinemia: Acetylcysteine initially 140 mg/kg then 70 mg/kg, ascorbic acid 10% 30 mg/kg, S-adenosyl methionine (Samyline). ** after a lack of response methylene blue at a dose of 1 mg/kg body weight was administered intravenously. *** No further treatment was needed after initial treatment. **** Reference values from Clinical Laboratory, Vetsuisse Faculty Zurich (Sysmex) or # from [35] Values highlighted in red are out of the reference range. MCV: mean cell volume, MCHC: mean corpuscular hemoglobin concentration. GSH: reduced glutathione.

**Table 2 cells-12-00991-t002:** Erythrocytic hydration and osmotic fragility parameters of a cat with hereditary CYB5R3 deficiency during steady-state conditions.

Parameters #	Control Cats (*n* = 5)	Affected Cat
EI_min_, AU	0.131 ± 0.033	0.142 ± 0.008
O_min_, mOsm	187.8 ± 5.6	186.3 ± 2.5
EI_max_, AU	0.517 ± 0.052	0.547 ± 0.002 **
O_EImax_, mOsm	359.4 ± 18.9	355.3 ± 2.5
EI_hyper_, AU	0.259 ± 0.027	0.274 ± 0.001 **
O_hyper_, mOsm	484.6 ± 7.6	462.0 ± 2.0 **
Area, AU	124.9 ± 14.2	118.4 ± 0.25
Na, mmol/kg dw	269.9 ± 13.7	264.8
K^+^, mmol/kg dw	13.7 ± 2.9	12.6

Means ± SD values from affected cat on three occasions (0.5, 4, and 6 months after surgery) and from five control cats. Ion content was measured on one occasion. # osmotic tolerance (area), deformability as maximal elongation index (EI_max_), osmolarity at which EI_max_ is reached (O_EImax_), hydration state as the half-maximal tolerated hyperosmotic osmolarity (O_hyper_), the elongation index at O_hyper_, minimal tolerated osmolarity (O_min_), elongation index at O_min_ (EI_min_), and dry weight (dw) [36]. ** Difference from the control (*p* < 0.05).

**Table 3 cells-12-00991-t003:** Variant filtering results of the affected cat against 74 control genomes.

Filtering Step	Heterozygous Variants	Homozygous Variants
Variants in the case genome	6,189,794	6,101,321
Private variants	90,816	23,992
Variants with SnpEff impact high, moderate or low	951	252
Private variants in *CYB5R3*	0	1

## Data Availability

The accessions for the sequence data reported in this study are listed in Appendix A.

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
