# Peer review of "Methemoglobinemia, Increased Deformability and Reduced Membrane Stability of Red Blood Cells in a Cat with a CYB5R3 Splice Defect"

_cells, 2023, doi:10.3390/cells12070991_

Round 1
Reviewer 1 Report
This is a well-written paper concerning a cat with hereditary methemoglobinemia. In addition to routine measurements, a thorough examination of the nature of the genetic defect, and some specialized assays were done to evaluate erythrocyte hydration, rheology, and internal calcium. A general concern is that some of the findings are emphasized more than seems appropriate and there is excess speculation. The microcytosis is slight, echinocytes are often found in cat blood samples, erythrocyte hydration and rheology values are only slightly different from controls, and it is not clear that these alterations are caused by the methemoglobinemia, as seems to be implied. With such mild changes, I would remove terms “dehydrated erythrocytes” from the title.
Line 27. Include the word mild before microcytosis.
Lines 28-29. Statement that erythrocytes were dehydrated needs to qualification. Data showed only slight differences from controls.
Lines 30-32. While this cat may have had increased erythrocyte fragility, deformability, and increase reduced glutathione (What is the basis to call it overload?), they are not proven to be the result of methemoglobinemia.
Lines 43-45. Another cause of hereditary methemoglobinemia is erythrocyte FAD deficiency - Harvey, JW, et. al. (2003) Methemoglobinemia and eccentrocytosis in equine erythrocyte flavin adenine dinucleotide deficiency. Vet Pathol 40: 632-642.
Line 31 – How are the erythrocyte morphologic alterations in this cat typical of stress erythropoiesis? This term generally refers to increased bone marrow response to an anemia. The reticulocyte count is not increased and erythrocytes tended to be small rather than large, as can be seen with stress erythropoiesis.
Lines 58-59. Heinz bodies were not described in cats in reference 11. Heinz bodies are present in normal cats at low levels and increased in some disorders such as diabetes and hyperthyroidism. Although Heinz bodies were reported in the case in reference 22, they were not likely associated with methemoglobinemia. I am not aware of evidence for increased oxidant generation in hereditary methemoglobinemia.
Lines 88-89. Please indicate what method was used to remove leukocytes and platelets before erythrocytes were washed and lysed. Were hemolysates centrifuged to remove erythrocyte membranes? These points are relative to the somewhat higher than expected enzyme activity in the patient.
Line 163. Was the first CBC done while the cat was under anesthesia, which might contribute to the lower initial HCT.
Table 1. Were CBCs done on additional days? Methylene blue can cause Heinz body formation in cats, although a single 1 mg/kg dose would not likely cause anemia. However, it is always prudent to check the HCT within a couple of days following methylene blue treatment in cats.
Lines 302-304. This is an overstatement that indicates a cause-and-effect relationship between this enzyme deficiency and microcytosis, dehydration, and high GSH values.
Line 374 – It may be just nomenclature, but GSH “overload” sounds like a pathologic accumulation such as iron overload.
Lines 375-376. Is there evidence for increased oxidative reaction resulting from increase methemoglobin as opposed to oxyHb and deoxyHb?
Lines 381-382. Is this a sign of stress erythropoiesis? It may be evidence of aged or damaged cells as in sickle cell anemia (two references given). Calcium accumulates in erythrocytes undergoing eryptosis.
Lines 403-404. Reference indicates that both methemoglobin oxidation AND membrane peroxidation are needed. I am not convinced that the presence of methemoglobin alone is sufficient.
Lines 408-409. Addressed earlier. Was their more than one case?
Lines 416-417. It is not the acquired methemoglobinemia that leads to intravascular hemolysis (nitrite toxicity results in methemoglobinemia without anemia). It is the oxidant damage to membranes and/or oxidized globin forming Heinz bodies that bind to and alter the membrane, resulting in anemia.
Author Response
We thank the reviewer for the constructive criticisms and suggestions. Point-to-point answers to the reviewer's comments are presented in a separate file.

Reviewer 2 Report
Comments on manuscript entitle “Methemoglobinemia and Dehydrated Erythrocytes in a cat with a CYB5R3 Splice Defect” by Sophia Jenni et al.
Manuscript ID: cells-2249409 - Review Report
1. The manuscript describing the Methemoglobinemia and Dehydrated Erythrocytes in a cat with a CYB5R3 Splice Defect. The manuscript is well written and have a valid research question or hypothesis.
2. The major concerns with the manuscript are lack of review of literatures in the introduction section and the diagnostic purpose of methemoglobinemia whether cyanosis and any clinical signs.
3. The author has not described in details the discussion section mechanism of methemoglobinemia is associated with an increase in red blood cell fragility and deformability, glutathione overload and morphological alterations typical for stress erythropoiesis..
4. The author has not mentioned about the Hb Level, hematocrit level and MCV, MCH level measured before and after the 1 mg/kg methylene blue injection in the table-1. It would be great if the authors do mention the same in the manuscript.
5. The author has described that NGS analysis is a useful tool to better diagnose patients with both metehemoglobnemia, But the author has mentioned in details the clinically cyanotic so no need for going for NGS used for genetic analysis. Sanger sequencing is sufficient to screen CYB5R gene mutation causes congenital methemoglobinemia.
6. The animal ethic approval statement to be included.
7. It would recommend the authors to elaborate the introduction part citing the recent literature with respect to methemoglobinemia caused by defects in CYB5R3 gene in human.
8. It would be important to show key data showing the efficacy of the targeted resequencing strategy, such as the threshold below which variants were discounted because of low coverage. It would also be useful to see coverage of the target genes offered.
9. It would be great to use in silico tools for predicting the pathogenicity of variants and also mention PDB code used for protein modelling .
10. I would also suggest the authors to check the manuscript for grammmarly corrections.
Author Response
We thank the reviewer for the constructive criticism. Please, find the point-to-point answers to the reviewer's comments in a separate file.

Round 2
Reviewer 1 Report
Spelling line 65 pyknocytosis and elsewhere, not piknocytosis
Author Response
We thank the reviewer for noticing this typo. It is corrected in the revised version.
Reviewer 2 Report
I am satisfied with the answer to my query.
Author Response
We thank the reviewer for helping us to improve the manuscript.